# Fangchinoline induces gallbladder cancer cell apoptosis by suppressing PI3K/Akt/XIAP axis

**Jiandong Li**[1,2,3☺], **Wenda Cen**[2,3,4☺], **Chenhao Tong**[1,2,3], **Luna Wang**[1,2,3], **Weiguang Zhang**[5], **Shiqing Deng**[2,3,4], **Jianhua Yu**[2,3,4]*, **Baochun Lu**[2,3,4]*

1 Zhejiang University School of Medicine, Hangzhou, Zhejiang, China, 2 Department of Hepatobiliary Surgery, Shaoxing People's Hospital (Shaoxing Hospital, Zhejiang University School of Medicine), Shaoxing, Zhejiang, China, 3 Shaoxing Key Laboratory of Minimally Invasive Abdominal Surgery and Precise Treatment of Tumor, Shaoxing, Zhejiang, China, 4 Shaoxing University Affiliated First Hospital, Shaoxing, Zhejiang, China, 5 Department of Molecular Medicine and Clinical Laboratory, Shaoxing Second Hospital, Shaoxing, Zhejiang, China

☺ These authors contributed equally to this work.
* yujianhua@zju.edu.cn (JY); sygd_lbc@126.com (BL)

**Data Availability Statement:** All relevant data are within the manuscript and its Supporting Information files.

**Funding:** This work was supported by Zhejiang Provincial Natural Science Foundation of China

## Abstract

Gallbladder cancer (GBC) is the most common biliary tract malignancy with a dismal prognosis. The development of new drugs may help to improve prognosis. This study found that fangchinoline, a bisbenzylisoquinoline alkaloids, inhibited the proliferation and clone formation of GBC cells in a dose-dependent manner. Moreover, Hoechst staining, TUNEL assays, and flow cytometry demonstrated that fangchinoline effectively induced apoptosis in GBC cells. Further studies found that an anti-apoptotic pathway, the PI3K/Akt/XIAP axis, was significantly inhibited in GBC cells after treating with fangchinoline. Finally, we confirmed that fangchinoline restrained xenograft tumor growth in vivo. Our findings indicate that fangchinoline can be considered a potential drug for GBC treatment.

## Introduction

Gallbladder cancer (GBC) is a highly lethal hepatobiliary malignancy originating from gallbladder mucosa and is the most common biliary tract tumor [1]. Notably, due to a lack of specific early detection methods and the asymptomatic nature of an initial stage, 40%-70% of GBC patients are diagnosed at an advanced stage with metastatic lesions [2], contributing to disease's dismal prognosis. According to National Cancer Data Base of the American College of Surgeons, the 5-year survival rates for GBC patients are separately 4% and 2% for stage IVa and stage IVb [2]. Consequently, the treatment of gallbladder cancer becomes extremely critical.

Surgical resection is the most effective treatment for GBC [3]. Unfortunately, many patients are always diagnosed at advanced stages, when surgery is no longer a possibility. While the standard first-line chemotherapeutic regimen of gemcitabine combined with cisplatin is required, its efficacy remains unsatisfactory [4]. Although immunosuppressants and molecularly targeted drugs have recently garnered increased attention [5, 6], they are expensive, and

under grant no. LY22H160008, LY19H160016, National Natural Science Foundation of China (NSFC) no. 81602044, Zhejiang Provincial Medical and Health Science and Technology Project under grant no. 2022RC080 and 2020RC127. Dr. Jianhua Yu received these funding and was responsible for the decision to publish this manuscript. This work was also supported by Zhejiang Provincial Natural Science Foundation of China under grant no. LBY22H030001 and Zhejiang Provincial Medical and Health Science and Technology Project under grant no. 2019ZD057. Dr. Baochun Lu received the funding and was responsible for data collection of this manuscript.

**Competing interests:** The authors have declared that no competing interests exist.

their curative effect appears to be ambiguous. It is necessary to identify an effective drug that can improve prognosis and reduce economic expenditure associated with GBC.

Fangchinoline (FAN), a bisbenzylisoquinoline alkaloid, is one of the main bioactive ingredients of Stephania tetrandra S. Moore and has been illustrated to hold potent pharmacological effects such as anti-inflammatory, antioxidant, and neuroprotective properties [7–9]. Notably, what makes this compound stand out is its anti-tumor activity in various cancers. Numerous researchers have recently discovered that fangchinoline can help alleviate the malignant phenotype of various tumors, including breast and lung cancer [10, 11]. Moreover, fangchinoline induces apoptosis in tumor cells mediated by various signaling pathways [11–13]. Several studies revealed that fangchinoline is regarded as a potential anticancer drug. However, no effective research regarding fangchinoline function on GBC has been conducted previously.

This study aimed to evaluate the role of fangchinoline on GBC cell lines GBC-SD and NOZ. Furthermore, the drug's effectiveness was demonstrated in vivo through tumor formation tests, and finally, the drug's possible underlying mechanism of action was clarified.

## Materials and methods

### Reagents

Fangchinoline (cas. no. 436-77-1) was purchased from Biopurify Phytochemicals Ltd. (Chengdu, China). SC79 (cat. no. SF2730) was obtained from Beyotime Institute of Biotechnology (Nanjing, China). Primary antibodies against FAK (cat. no. 3285), Phospho-FAK (Tyr576/577, cat. no. 3281), Src (cat. no. 2123), Phospho-Src (Tyr527, cat. no. 2105), PI3K (cat. no. 4249), Phospho-PI3K p85 (Tyr458)/p55 (Tyr199) (cat. no. 4228), Phospho-Akt (Ser473, cat. no. 4060), mTOR (cat. no. 2972), Phospho-mTOR (Ser2448) (cat. no. 5536), XIAP (cat. no. 2045), and β-Actin (cat. no. 3700) were purchased from Cell Signaling Technology (Boston, MA, USA). Primary antibodies against Akt (cat. no. 10176-2-AP) were purchased from Proteintech (Wuhan, China). HRP-conjugated secondary antibodies (cat. nos. A0216 and A0208), PMSF (cat. no. ST506), Enhanced BCA Protein Assay kit (cat. no. P0010), RIPA Lysis Buffer (cat. no. P0013B), Hoechst Staining Kit (cat. no. C0003), One Step TUNEL Apoptosis Assay Kit (cat. no. C1089) were obtained from Beyotime Institute of Biotechnology. CCK-8 was obtained from MCE (Monmouth Junction, NJ, USA). Annexin V Apoptosis Detection Kit was gained from BD Biosciences (Franklin Lakes, NJ, USA).

### Cell culture

The human GBC cell line GBC-SD was acquired from the Chinese Academy of Science Shanghai Branch Cell Bank (Shanghai, China). NOZ cell line was purchased from the HAKATA Cell Bank of the Shanghai Chuanqiu Biotechnology Co., LTD (Shanghai, China). GBC-SD cells were cultured in RPMI-1640 and NOZ cells were cultured in DMEM medium with 10% fetal bovine serum (FBS) in the incubator at an atmosphere of 37˚C containing 5% $CO_2$. Fangchinoline was dissolved in DMSO and diluted in medium (with the same level of DMSO in the control group) to a final value of DMSO<0.1%.

### Cell viability assays

Cells were plated in 96-well plates at a density of $4\times10^3$ cells per well and incubated in medium containing different concentrations of fangchinoline for 48 h. The plates were incubated at 37˚C for another 2 h after one-tenth volume of CCK8 reagent added to each well. Cell density was determined by measuring the absorbance at 450 nm with a microplate reader.

## Colony formation assays

After cell counting, GBC cells were seeded in 6-well plates at $5 \times 10^2$ cells per well and cultured overnight. The GBC cells were treated with different concentrations of fangchinoline, while the culture medium containing DMSO was used as the control group. After 14 days, the cells were fixed with 4% paraformaldehyde and then stained with 0.01% crystal violet.

## Cell apoptosis analysis

GBC-SD cells were incubated with different concentrations of fangchinoline for 48 h. After treating with EDTA-free trypsin, the cells were collected and resuspended in 1×Binding Buffer sufficiently. The resuspended cells were stained with Annexin V-FITC and propidium iodide (PI) (Becton-Dickinson, Franklin, NJ, USA) following the manufacturer's instructions. The apoptosis rate was determined by flow cytometry (Beckman Coulter, Fullerton, CA, USA).

## Hoechst 33258 staining

GBC cells were seeded in 24-well plates at $2 \times 10^4$ cells per well and cultured overnight. Then, the GBC cells were cultured with different concentrations of fangchinoline for further 48 h. The concentrated nuclei were observed under a Nikon fluorescence microscope (Tokyo, Japan) according to the instructions of the apoptosis staining kit (cat. no. C0003; Beyotime).

## TUNEL assays

Briefly, GBC cells were plated in 96-well plates. The culture condition was changed to the serum-free medium containing DMSO or fanchinoline when the density of cells was about 60%. After 48 h, the cells were fixed, permeated, and then stained with a TUNEL kit (cat. no. C1089; Beyotime) according to manufacturer's instructions.

## Treating cells with SC79

The GBC cells were seeded in 6-well plates at about 40% densities and cultured overnight. After starving with serum-free medium for 1 h, the GBC cells were treated with medium with 5% FBS plus 4μg/ml SC79 for 30 min. After incubated with different concentrations of fang-chinoline for 48 h, the cells were harvested for western blot analysis.

## Western blot analysis

Total protein extracted from GBC cells was obtained with RIPA lysis (cat. no. P0013B; Beyo-time) buffer containing 1% PMSF (cat. no. ST506; Beyotime). After quantified by BCA kit (cat. no. P0010; Beyotime), 30 μg protein sample was electrophoresed on a 10% SDS–PAGE gel and transferred onto a polyvinylidene fluoride membrane. The membrane was blocked and incubated with a primary antibody, followed by incubation with a horseradish peroxidase-conjugated secondary antibody (cat. nos. A0216 and A0208; Beyotime). Immunoreactive bands were visualized using a chemiluminescence solution (Millipore, Temecula, CA, USA), β-Actin was used as the endogenous control.

## Xenograft formation assays

The six-week-old male athymic nude mice were purchased from SLAC Laboratory Animal Company (Shanghai, China). To establish a tumor xenograft model, the backs of all nude mice were injected subcutaneously with 200μl 0.9% normal saline containing GBC-SD cells ($2 \times 10^6$). After a week, the nude mice with tumor xenograft were randomly divided into two groups

(n = 7). The experimental group was intraperitoneally injected with fangchinoline (5mg/kg, dissolved in 20% DMSO) while the control group was injected with 0.9% normal saline (20% DMSO) every other day for 3 weeks. To minimize the suffering, all mice were sacrificed via cervical dislocation under sevoflurane inhalation anesthesia after four weeks, the death of the mice was confirmed by examining the cessation of vital signs. The weight and volume of the tumors were measured and the tumor volume was calculated by the following formula: volume = length$\times$width$^2$/2. All the mice had ad libitum access to food and water and were maintained at an atmosphere of 20–25˚C and 50–60% humidity under a cycle of 12 h light/dark. All procedures were approved by the Ethics Committee of Shaoxing People's Hospital and conformed to the ARRIVE guidelines 2.0 published in PLOS Biology.

## Statistical analysis

All experiments were repeated at least three times, and data are presented as the means ± SD. Student's t-test was used to determine statistical significance between two groups. One-way ANOVA followed by the Tukey–Kramer adjustment was used to examine differences among multiple groups. All statistical analyses were conducted using SPSS v21.0 (IBM, Armonk, NY, USA), and $P < 0.05$ was considered statistically significant.

## Results

### Fangchinoline inhibits the proliferation of GBC cells

CCK8 assays were used to determine the effect of fangchinoline on the proliferation of GBC cells. The concentration-dependent cell viability curves revealed that fangchinoline inhibited the proliferation of GBC-SD and NOZ cells in a dose-dependent manner (Fig 1A). Additionally, GBC-SD cells were more sensitive to fangchinoline compared with NOZ cells. Colony formation assay also revealed that fangchinoline significantly inhibited the colony-forming ability of GBC cells, compared with the control group (Fig 1B).

### Fangchinoline induces the apoptosis of GBC cells

Various anticancer drugs could inhibit proliferation by inducing apoptosis of cancer cells as the more common mechanism [14–16]. To examine whether fangchinoline induces tumor apoptosis, Hoechst staining, TUNEL assays, and Annexin V/PI apoptosis assays were performed. Hoechst staining revealed that after fangchinoline treatment, more GBC-SD and NOZ cells had a concentrated nuclear population (Fig 2A). TUNEL staining also revealed that the fangchinoline-treated group had more TUNEL-positive cells than the control group (Fig 2B). Moreover, Annexin V/PI assays confirmed that fangchinoline-treated GBC-SD cells had a higher proportion of apoptotic cells and a lower proportion of live cells than the untreated group (Fig 2C). Overall, sufficient data demonstrated that fangchinoline induced apoptosis of GBC cells.

### Fangchinoline inhibits the PI3K/Akt/XIAP axis

PI3K/Akt is an important cancer-related pathway that influences cancer cell apoptosis and survival [17, 18]. Western blot was conducted to investigate the impact of fangchinoline on PI3K/Akt pathway in GBC-SD and NOZ cells. The results indicated that fangchinoline could inhibit PI3K and Akt expression (Fig 3A). XIAP is a PI3K/Akt pathway downstream protein that inhibits cell apoptosis [19, 20]. Additionally, we discovered that XIAP was dose-dependently downregulated (Fig 3A).

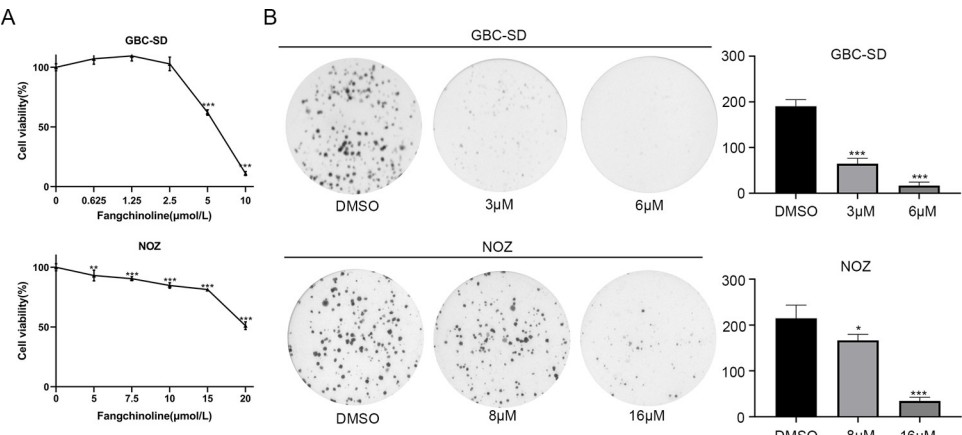

**Fig 1. Fangchinoline suppresses the proliferation of GBC cells.** (A) The cell viability curves detected using CCK8 assays after treating GBC-SD and NOZ cells with different fangchinoline concentrations. (B) The colony formation ability of GBC-SD and NOZ cells treated with different fangchinoline concentrations. Data are presented as mean ± SD. *$P<0.05$. **$P<0.01$. ***$P<0.001$, compared with the DMSO group.

To further confirm the interaction between p-Akt and XIAP, an agonist of Akt pathway, SC79, was employed to upregulate p-Akt expression. The expression level of XIAP was markedly upregulated, consistent with p-Akt after treating with SC79 (Fig 3B). Interestingly, fangchinoline treatment effectively reversed SC79-induced increases in XIAP and p-Akt expression (Fig 3B). It indicated that inhibiting PI3K/Akt/XIAP axis was the potential mechanism of how fangchinoline induced GBC cell apoptosis.

## Fangchinoline suppresses the oncogenesis of GBC-SD cells in vivo

A xenograft formation assay in nude mice was performed to further evaluate fangchinoline's potential efficacy in vivo. The results revealed that the weight and volume of tumors in the fangchinoline-treated group were decreased compared with those in the control group (Fig 4; Table 1). Moreover, none of tumor-bearing mice died during the experiment, demonstrating

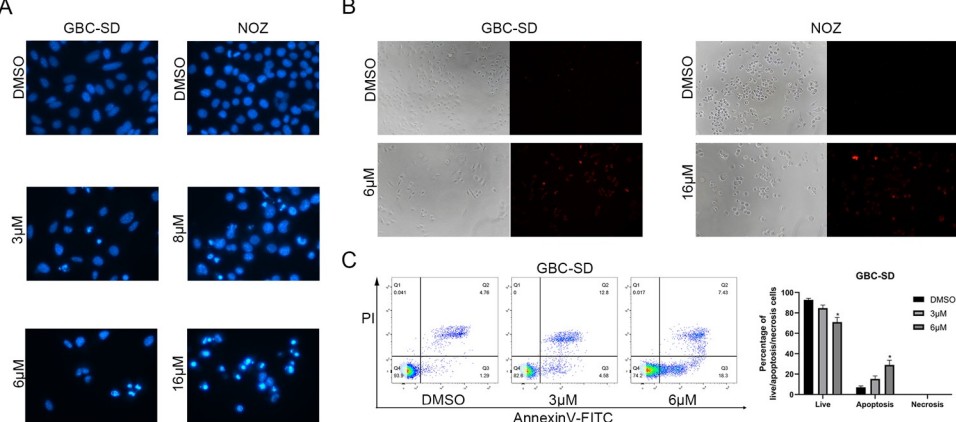

**Fig 2. Fangchinoline induces the apoptosis of GBC cells.** (A) Hoechst 33258 staining of GBC cells treated with fangchinoline. (B) TUNEL assay of the apoptosis effect of GBC cells with high fangchinoline concentration. (C) Annexin V/ PI apoptosis assay of GBC-SD. Annexin V (+)/PI (−) or Annexin V (−)/PI (+) were considered as apoptotic cells, Annexin V (−)/PI (−) cells were alive, and Annexin V (+)/PI (+) cells were necrotic.

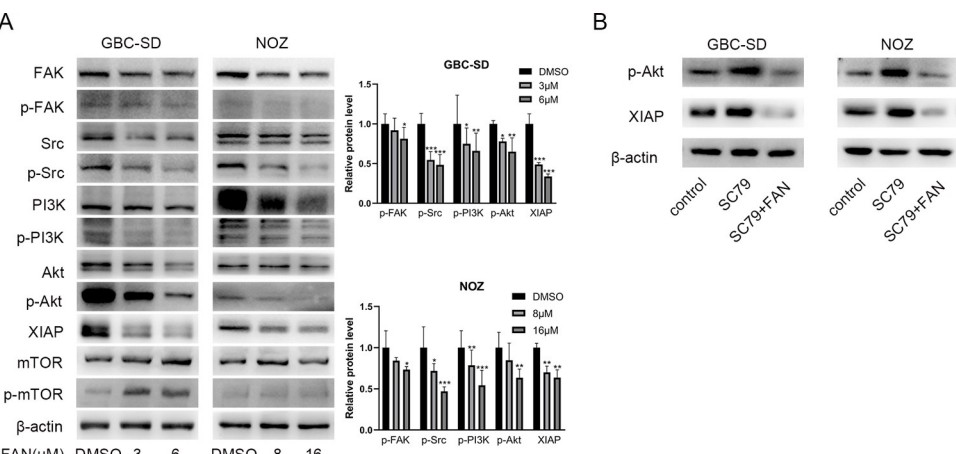

**Fig 3. Fangchinoline inhibits the PI3K/Akt/XIAP signaling axis.** (A) Results of FAK, p-FAK (Tyr576/577), Src, p-Src (Tyr527), PI3K, p-PI3K (Tyr458/Tyr199), AKT, p-AKT (Ser473), mTOR, p-mTOR (Ser2448), and XIAP relative protein expression using Western blot. (B) GBC cells were incubated with p-Akt activator SC-79 alone or combined with fangchinoline before lysis for Western blot. Data are presented as mean ± SD. $^*P<0.05$. $^{**}P<0.01$ and $^{***}P<0.001$, compared with the DMSO group.

the feasibility of fangchinoline effectively inhibiting tumor progression. It indicated that fangchinoline effectively suppressed the oncogenesis of GBC-SD cells in vivo.

## Discussion

Apoptosis, also known as programmed cell death, is a crucial self-stabilizing mechanism in multicellular organisms [21]. It eliminates incompatible cells from the body to maintain a healthy balance [22–24]. Numerous studies indicate that diseases such as cancer may be linked to underlying defects in apoptosis pathways which keep cells alive that should die [25, 26].

Fangchinoline has been shown to have anti-tumor effects, inhibiting the formation and development of multiple tumor cells. Previous study reported that fangchinoline promotes apoptosis in prostate cancer cells by suppressing the proteasome β1 subunit [27]. Similarly, fangchinoline has been demonstrated to promote apoptosis in human pancreatic cancer cells [12], breast cancer cells [10], and bladder cancer cells [28]. Although the mechanism of action is various, these studies indicated that fangchinoline functions as an anti-tumor drug in multiple tumor types by inducing apoptosis. Through various experimental methods, we demonstrated here that fangchinoline also effectively induces apoptosis in GBC cells.

Focal adhesion kinase (FAK) is a cytoplasmic tyrosine kinase lacking a transmembrane region [29]. Numerous studies have established that it is critical in the progression of tumors toward malignant and invasive phenotypes [30, 31]. Fangchinoline has been demonstrated to inhibit FAK-related signaling pathway. After activation, FAK can be tightly bound to Src

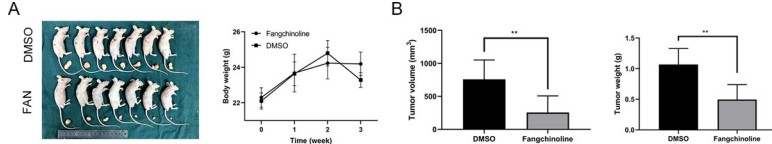

**Fig 4. Fangchinoline suppresses the oncogenesis of GBC-SD cells in vivo.** (A) The photograph of nude mice and xenograft tumors. (B) The volume and weight of xenograft tumors were measured. $^{**}P < 0.01$, compared with the DMSO group.

**Table 1. Tumor growth in nude mice.**

| Group | n | Volume (mm$^3$) | Weight (g) |
|---|---|---|---|
| DMSO | 7 | 758.14±293.18† | 1.07±0.26 |
| Fangchinoline | 7 | 256.89±250.61 | 0.50±0.24 |

† Mean.

± Standard Deviation.

family genes via SH2 domain, activating downstream to control cell growth [32, 33]. Additionally, we examined FAK-related signaling pathway in this study. However, we discovered that fangchinoline had no significant effect on FAK expression. Therefore, PI3K/Akt signaling pathway associated with FAK attracted our attention.

PI3K, a member of lipid kinase family, phosphorylates the 3'-OH group of inositol ring to generate the second messenger PIP3 in the plasma membrane [17]. PI3K successfully recruits a subset of signaling proteins such as Akt with pleckstrin homology (PH) domain to mediate cell proliferation and apoptosis by virtue of PIP3 [34, 35]. In the whole cascade, Akt is considered the central mediator of PI3K/Akt signaling pathway. Fangchinoline can promote the apoptosis of human glioblastoma cells by inhibiting Bax and caspase9 expression through Akt pathway [36]. Similar mechanisms have been observed in breast cancer cells [37] and leukemia cells [38]. Consistent with previous findings, we discovered that fangchinoline could significantly downregulate the expression of PI3K, Akt, and associated phosphorylated proteins. In osteosarcoma, fangchinoline suppresses the migration and invasion of cells through the PI3K/Akt pathway [39]. And apoptosis is induced through activating downstream caspase which is different from our results. In our experiments, fangchinoline had no significant effect on metastasis of gallbladder cancer cells. XIAP is an intracellular anti-apoptotic protein in GBC-SD and NOZ cells. The expression of XIAP was significantly downregulated after drug treatment. A study reported that Akt phosphorylation could inhibit XIAP ubiquitination, reduce its degradation, and partially reverse cisplatin-induced apoptosis of ovarian cancer epithelial cells [40], indicating that XIAP can be an Akt substrate. Our experiments unambiguously demonstrated an interaction between p-Akt and XIAP. Therefore, it is reasonable to conclude that fangchinoline can promote the apoptosis of GBC cells by inhibiting XIAP expression via PI3K/Akt pathway.

In summary, this study mainly probed the effects of fangchinoline on GBC-SD and NOZ gallbladder cancer cells. The results demonstrated that fangchinoline induced the apoptosis of GBC cells by regulating PI3K/Akt/XIAP signaling axis.

## Supporting information

**S1 File.**
(ZIP)

## Author Contributions

**Data curation:** Shiqing Deng.

**Formal analysis:** Baochun Lu.

**Methodology:** Weiguang Zhang.

**Project administration:** Luna Wang.

**Software:** Chenhao Tong.

**Writing – original draft:** Jiandong Li, Wenda Cen.

**Writing – review & editing:** Jianhua Yu.

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
