## [Decision Letter · Decision Letter 0]

18 Jan 2022

PONE-D-21-34482Fangchinoline induces gallbladder cancer cell apoptosis by suppressing PI3K/Akt/XIAP axisPLOS ONE

Dear Dr. Yu,

Thank you for submitting your manuscript to PLOS ONE. After careful consideration, we feel that it has merit but does not fully meet PLOS ONE’s publication criteria as it currently stands. Therefore, we invite you to submit a revised version of the manuscript that addresses the points raised during the review process.

 Regarding blot/gel data: PLOS ONE now requires that submissions reporting blots or gels include original, uncropped blot/gel image data as a supplement or in a public repository as described at https://journals.plos.org/plosone/s/figures#loc-blot-and-gel-reporting-requirements. These requirements apply both to the main figures and to cropped blot/gel images included in Supporting Information.

We look forward to receiving your revised manuscript.

Kind regards,

Irina V. Balalaeva, PhD

Academic Editor

PLOS ONE

Journal Requirements:

2. To comply with PLOS ONE submissions requirements, in your Methods section, please provide additional information on the animal research and ensure you have included details on (1) methods of sacrifice, (2) methods of anesthesia and/or analgesia, (3) efforts to alleviate suffering, (4) number of animals used in the study.

“The work was sponsored by Zhejiang Provincial Natural Science Foundation of China under grant no. LY19H160016, National Natural Science Foundation of China (NSFC) no. 81602044, Zhejiang Provincial Medical and Health Science and Technology Project under grant no. 2019ZD057.”

“This work was supported by National Natural Science Foundation of China (NSFC) no. 81602044 and Zhejiang Provincial Natural Science Foundation of China under grant no. LY19H160016. Dr. Jianhua Yu received these funding and was responsible for the decision to publish this manuscript.

This work was also supported by Zhejiang Provincial Medical and Health Science and Technology Project under grant no. 2019ZD057. Dr. Baochun Lu received the funding and was responsible for data collection of this manuscript.”

Reviewers' comments:

Reviewer's Responses to Questions

**Comments to the Author**

1. Is the manuscript technically sound, and do the data support the conclusions?

Reviewer #1: Yes

Reviewer #2: Partly

2. Has the statistical analysis been performed appropriately and rigorously? 

Reviewer #1: Yes

Reviewer #2: Yes

3. Have the authors made all data underlying the findings in their manuscript fully available?

Reviewer #1: Yes

Reviewer #2: No

4. Is the manuscript presented in an intelligible fashion and written in standard English?

Reviewer #1: Yes

Reviewer #2: No

5. Review Comments to the Author

Reviewer #1: I consider that this article contains important results on the effect of fangchinoline (a bisbenzylisoquinoline alkaloid) wher the authors demonstrated the inhibition of the proliferation and clone formation of GBC cells in a dose-dependent manner and the induction of apoptosis in GBC cells. Authors also showed that fangchinoline acts in vivo by restraining xenograft tumor growth on six-week-old male athymic nude mice.

The document is written correctly and there is a good review of the literature on the subject developed, in addition to meeting the criteria of the journal, so I consider it can be published without problem.

Reviewer #2: The manucript Fangchinoline induces gallbladder cancer cell apoptosis by suppressing PI3K/Akt/XIAP

axis lacks the selection of this alkaloid as an inhibitor of PI3K/Akt/XIAP? why not PI3K/Akt/mTOR?

Also, the rationality is required to show the anti-cancer activity of Fangchinoline.

The author should use positive control in all the assays.

Fangchinoline suppresses the proliferation, invasion and tumorigenesis of human osteosarcoma cells through the inhibition of PI3K and downstream signaling pathways is reported IJMM 2017.

So better the author compare some results with the published mode of action of this alkaloid and resubmit.

6. PLOS authors have the option to publish the peer review history of their article (what does this mean?). If published, this will include your full peer review and any attached files.

Reviewer #1: No

Reviewer #2: **Yes: **BASAPPA BASAPPA

---

## [Author Response · Author response to Decision Letter 0]

15 Feb 2022

Responses to the comments of Reviewer #1

I consider that this article contains important results on the effect of fangchinoline (a bisbenzylisoquinoline alkaloid) wher the authors demonstrated the inhibition of the proliferation and clone formation of GBC cells in a dose-dependent manner and the induction of apoptosis in GBC cells. Authors also showed that fangchinoline acts in vivo by restraining xenograft tumor growth on six-week-old male athymic nude mice.

The document is written correctly and there is a good review of the literature on the subject developed, in addition to meeting the criteria of the journal, so I consider it can be published without problem.

Response: Thank you for the Reviewer’s positive comments.

Responses to the comments of Reviewer #2

1. The manucript Fangchinoline induces gallbladder cancer cell apoptosis by suppressing PI3K/Akt/XIAP axis lacks the selection of this alkaloid as an inhibitor of PI3K/Akt/XIAP? why not PI3K/Akt/mTOR?

Response: As the comments of Reviewer, we have examined the PI3K/Akt/mTOR axis and provided the results in the revised manuscript (Fig3). Unfortunately, the difference of p-mTOR was not significant with or without Fangchinoline treatment (Fig3). On the other hand, the relationship between XIAP and Akt had been further confirmed by an Akt activator, SC79 (Fig3). 

2. Also, the rationality is required to show the anti-cancer activity of Fangchinoline. The author should use positive control in all the assays.

Response: The suggestion from the reviewer is helpful. However, the aim of our study is to determine whether Fangchinoline effectively inhibits GBC proliferation in vivo and in vitro, rather than to compare it with other anti-cancer drugs. Similar research strategy has been used by abundant studies. [1-5]

3. Fangchinoline suppresses the proliferation, invasion and tumorigenesis of human osteosarcoma cells through the inhibition of PI3K and downstream signaling pathways is reported IJMM 2017. So better the author compare some results with the published mode of action of this alkaloid and resubmit.

Response: As the comments of Reviewer, the comparison of relevant conclusions has been added to our revised manuscript.

Responses to the comments of Editor

Response: As the comments of Editor, we have mad revisions in our revised manuscript, including style and file naming.

2. To comply with PLOS ONE submissions requirements, in your Methods section, please provide additional information on the animal research and ensure you have included details on (1) methods of sacrifice, (2) methods of anesthesia and/or analgesia, (3) efforts to alleviate suffering, (4) number of animals used in the study.

Response: As the comments of Editor, the additional information on the animal research has been added into the section of Xenograft formations assays.

“The work was sponsored by Zhejiang Provincial Natural Science Foundation of China under grant no. LY19H160016, National Natural Science Foundation of China (NSFC) no. 81602044, Zhejiang Provincial Medical and Health Science and Technology Project under grant no. 2019ZD057.”

“This work was supported by National Natural Science Foundation of China (NSFC) no. 81602044 and Zhejiang Provincial Natural Science Foundation of China under grant no. LY19H160016. Dr. Jianhua Yu received these funding and was responsible for the decision to publish this manuscript.

This work was also supported by Zhejiang Provincial Medical and Health Science and Technology Project under grant no. 2019ZD057. Dr. Baochun Lu received the funding and was responsible for data collection of this manuscript.”

Response: As the comments of Editor, the funding-related text in manuscript has been removed. Meanwhile, we have updated the funding information as the comments of Editor:

“This work was supported by Zhejiang Provincial Natural Science Foundation of China under grant no. LY22H160008, LY19H160016, National Natural Science Foundation of China (NSFC) no. 81602044, Zhejiang Provincial Medical and Health Science and Technology Project under grant no. 2022RC080 and 2020RC127. Dr. Jianhua Yu received these funding and was responsible for the decision to publish this manuscript.

This work was also supported by Zhejiang Provincial Natural Science Foundation of China under grant no. LBY22H030001 and Zhejiang Provincial Medical and Health Science and Technology Project under grant no. 2019ZD057. Dr. Baochun Lu received the funding and was responsible for data collection of this manuscript.”

Thanks to the Editors for changing the online submission form on our behalf.

4. In your Data Availability statement, you have not specified where the minimal data set underlying the results described in your manuscript can be found.

Response: As the comments of Editor, the minimal data set underlying results has been added into our supplementary materials. All relevant data now are within the revised manuscript and its Supporting information files. 

5. PLOS ONE now requires that authors provide the original uncropped and unadjusted images underlying all blot or gel results reported in a submission’s figures or Supporting Information files.

Response: As the comments of Editor, the original uncropped and unadjusted images underlying all blot or gel results have been uploaded and named ‘S1_raw_images’.

Response: We have reviewed our reference list as the Reviewer suggested and there are no papers have been retracted. It’s worth noting that a new papers[6] was cited in our Discussion section.

---

## [Editor Report · Decision Letter 1]

28 Mar 2022

Fangchinoline induces gallbladder cancer cell apoptosis by suppressing PI3K/Akt/XIAP axis

PONE-D-21-34482R1

Dear Dr. Yu,

We’re pleased to inform you that your manuscript has been judged scientifically suitable for publication and will be formally accepted for publication once it meets all outstanding technical requirements.

Kind regards,

Irina V. Balalaeva, PhD

Academic Editor

PLOS ONE
---

## [Editor Report · Acceptance letter]

12 Apr 2022

PONE-D-21-34482R1 

Fangchinoline induces gallbladder cancer cell apoptosis by suppressing PI3K/Akt/XIAP axis 

Dear Dr. Yu:

I'm pleased to inform you that your manuscript has been deemed suitable for publication in PLOS ONE. Congratulations! Your manuscript is now with our production department. 

Kind regards, 

on behalf of

Dr. Irina V. Balalaeva 

Academic Editor

PLOS ONE